# The Builders of the Junction: Roles of Junctophilin1 and Junctophilin2 in the Assembly of the Sarcoplasmic Reticulum–Plasma Membrane Junctions in Striated Muscle

**DOI:** 10.3390/biom12010109

**Published:** 2022-01-10

**Authors:** Stefano Perni

**Affiliations:** Department of Physiology and Biophysics, Anschutz Medical Campus, University of Colorado, Aurora, CO 80045, USA; stefano.perni@cuanschutz.edu

**Keywords:** striated muscle, ER-PM junctions, junctophilins, excitation-contraction coupling

## Abstract

Contraction of striated muscle is triggered by a massive release of calcium from the sarcoplasmic reticulum (SR) into the cytoplasm. This intracellular calcium release is initiated by membrane depolarization, which is sensed by voltage-gated calcium channels Ca_V_1.1 (in skeletal muscle) and Ca_V_1.2 (in cardiac muscle) in the plasma membrane (PM), which in turn activate the calcium-releasing channel ryanodine receptor (RyR) embedded in the SR membrane. This cross-communication between channels in the PM and in the SR happens at specialized regions, the SR-PM junctions, where these two compartments come in close proximity. Junctophilin1 and Junctophilin2 are responsible for the formation and stabilization of SR-PM junctions in striated muscle and actively participate in the recruitment of the two essential players in intracellular calcium release, Ca_V_ and RyR. This short review focuses on the roles of junctophilins1 and 2 in the formation and organization of SR-PM junctions in skeletal and cardiac muscle and on the functional consequences of the absence or malfunction of these proteins in striated muscle in light of recently published data and recent advancements in protein structure prediction.

## 1. Introduction

Striated muscle evolved in early free-living invertebrates to confer locomotion to the individual and allow the search for food and the avoidance of predators or harmful environments. The term striated arises from the typical striated pattern of this tissue when observed in light and electron microscopy and defines both skeletal and cardiac muscle. This striation is the manifestation of the finely organized contractile apparatus in each of the functional contractile units of the muscle, the sarcomeres, which are arranged in series in the longitudinal direction of the muscle fiber. By the 1960s, it was suspected that a chemical activator, later identified as calcium ions (Ca^2+^), was responsible for muscle contraction, but it was still puzzling how a soluble activator, with a relatively slow diffusion speed, could be responsible for the fast and uniform contraction of a muscle fiber that can be tens of microns thick. Experiments conducted by Andrew Huxley and colleagues on frog, lizard and crab skeletal muscle revealed that the minimal electrical stimulus necessary to achieve local contractions in the muscle fiber was lower in areas of the fiber that fell along specific sections of the sarcomere [1,2]. Such sections corresponded to regions where particular structures, called triads, were observed in electron microscopy [3] (Figure 1).

Ultrastructural studies of the triad revealed a tri-partite organization, hence the name triad, in which two enlarged regions of the sarcoplasmic reticulum (SR), called terminal cisternae, sandwich another membrane structure, the T-tubule, which is in direct continuity with the plasma membrane (PM) [7,8]. The continuity of the T-tubule with the plasma membrane ensures fast transmission of the membrane depolarization into the interior of the fiber, allowing for a uniform contraction. Therefore, the triad in skeletal muscle represents a highly specialized form of ER-PM junction in which the SR membrane and the T-tubule are joined together in close proximity.

Mammalian ventricular cardiac muscle shows a similar organization to that of skeletal muscle with T-tubules and SR terminal cisternae. However, the specialized ER-PM junctions are arranged as dyads with a T-tubule contacting a single SR terminal cisterna [9] rather than triads (Figure 1). Mammalian atrial myocytes and cardiac muscle from non-mammalian vertebrates lack T-tubules, and the ER-PM junctions are in the form of peripheral couplings, where the SR is in direct contact with the plasma membrane at the periphery of the fiber [6,10,11,12,13].

Triads in skeletal muscle and dyads and peripheral couplings in cardiac muscle are the sites at which membrane depolarization is translated into Ca^2+^ release from the SR and, eventually, muscle contraction. This process, called excitation–contraction (EC) coupling, requires the cross-talk between two main players: the ryanodine receptor (RyR), a highly conductive Ca^2+^ channel embedded in the SR membrane and responsible for the rapid release of Ca^2+^ from the SR, and the L-type Ca^2+^ channel in the plasma membrane (T-tubule), which senses membrane depolarization and activates the RyR. The way RyR is activated differs in the cardiac and skeletal muscle systems. In the former, the opening of the cardiac muscle L-Type channel Ca_V_1.2 generates a rapid Ca^2+^ influx through the channel from the extracellular environment, causing a rapid increase in Ca^2+^ concentration in the narrow space separating the T-tubule and the SR in the dyad, and inducing the opening of the cardiac RyR isoform, RyR2. This mechanism is defined as calcium-induced calcium release (CICR) [14]. In skeletal muscle, the release of Ca^2+^ from the SR is directly triggered by the activation of the skeletal muscle L-type channel, Ca_V_1.1. The voltage-induced conformational change in Ca_V_1.1 is mechanically transmitted to RyR1 [15], causing its opening. This mechanism, which is independent of Ca^2+^ influx through Ca_V_1.1, is known as voltage-induced calcium release (VICR).

In addition to the voltage sensor in the membrane (Ca_V_) and the Ca^2+^-releasing channel in the SR (RyR), the voltage-gated Ca^2+^ channels β subunits are also essential for EC coupling [16,17]. Ca_V_1.1 and Ca_V_1.2 channels bind to the skeletal β1a and cardiac β2a, respectively, through their alpha interacting domain (AID) located in the intracellular loop connecting transmembrane domains I and II [18]. Beta subunits are crucial for facilitating the trafficking of the channel into the plasma membrane and for modulating the channel activity [16,19]. In skeletal muscle, the adapter protein Stac3 is also required for voltage-induced calcium release [20,21]. The exact role of Stac3 in skeletal muscle EC coupling is yet to be elucidated. Stac3 facilitates, but is not essential for, the membrane trafficking of Ca_V_1.1 [22,23]; nonetheless, knocking out Stac3 completely abolishes the voltage-induced Ca^2+^ release [20,21]. The observation that Stac3 binds to the II-III intracellular loop of Ca_V_1.1 [24], which is critical for the cross-talk between Ca_V_1.1 and RyR1 [25], suggests that Stac3 might allow or facilitate the mechanical coupling between these two channels.

It appears evident that the association of all the EC coupling essential and accessory proteins and the efficient cross-talk between Ca_V_1 in the plasma membrane and RyR in the SR require the accurate formation and organization of ER (SR)-PM junctions, making the proteins responsible for the organization of such junctions also essential for EC coupling. This short review focuses on the proteins responsible for the formation of SR-PM junctions in striated muscle, junctophilin1 and junctophilin2. Junctophilins’ physiological role in the structural and functional assembly of triads and dyads and the pathological consequences of their absence or malfunction in skeletal and cardiac muscle will be discussed. Finally, new developments in predicting the structure of junctophilins, and the possible implication for their functions will be touched on.

## 2. The Junctophilin Family

Junctophilins (JPH1, JPH2, JPH3 and JPH4) were discovered by Takeshima and collaborators in the early 2000s in muscles and neurons [26,27]. All four isoforms of this family contain a C-terminal transmembrane domain that embeds in the ER membrane and eight N-terminal domains called membrane occupation and recognition nexuses (MORN), thought to be responsible for the association with the plasma membrane (Figure 2).

This particular organization allows the junctophilins to link the ER membrane and the plasma membrane to one another. The MORN motifs are known to bind to the internal leaflet of the plasma membrane and specifically to phosphatidylinositol phosphate species [29,30,31], although the role of the MORN motif as a bona fide membrane-binding domain has been recently called into question after the observation that a number of proteins that also contain MORN repeats are either not associated with membranes or employ alternative domains to anchor to the membrane, or utilize their MORN motifs as protein-protein–protein interaction domains rather than membrane association domains [32,33,34]. Therefore, the ability of the MORN motifs in the junctophilins to associate with the membrane might arise from the presence of positively charged residues in the less conserved regions or from post-transcriptional modifications. Jiang and colleagues [35] have recently shown that palmitoylation at cysteine residues in MORN 1 and 8 of JPH2, which are largely conserved in other junctophilin isoforms, is involved in the formation of stable ER-PM junctions in CHO cells and is instrumental for the association of JPH2 with lipid rafts.

In addition to the MORN motifs and the transmembrane domain, junctophilins contain a mostly disordered region that connects the sixth and seventh MORN motifs, termed joining domain, a putative α-helical region following the eighth MORN domain and an extended disordered divergent region that separates the α-helical region and the C-terminal transmembrane domain (Figure 2). As suggested by its designation, the divergent domain is poorly conserved among different isoforms, while the rest of the sequence shows high (MORN motif and transmembrane domain), moderate (α-helical domain) or mild (joining domain) conservation among isoforms [27].

Junctophilin expression can be detected in all excitable cells, with different isoforms having tissue-specific expressions. JPH3 and JPH4 were first discovered in the brain, where they are expressed almost ubiquitously [26] and in sensory neurons [36]. Junctophilins 3 and 4 were also detected in other non-neuronal tissues such as pancreatic β-cells [37] and T-type lymphocytes [38]. Junctophilin 1 and 2 are mostly expressed in muscle tissues. In particular, Junctophilin 1 is expressed primarily in skeletal muscle. Junctophilin 2 is expressed in skeletal muscle and is the only isoform expressed in cardiac muscle [27]. Junctophilin 1 expression has also been found in the peripheral nervous system [39], and Junctophilin 2 is also highly expressed in smooth muscle [40].

While, in addition to junctophilins, neuronal cells express a variety of proteins that are able to form stable ER-PM junctions [41], JPH1 and JPH2 are the only proteins that form ER-PM junctions in striated muscle. The forces that keep these junctions together are considerably strong since they have to withstand mechanical stress provided by the repetitive contractions and stretching of the muscle fiber, which in some cases can reach extreme levels [42,43]. However, junctophilins’ role is not limited to building such strong structures; they also actively recruit and interact with the components that populate these SR-PM junctions and form the functional apparatus responsible for EC coupling.

## 3. Recruitment of Junctional Proteins by Junctophilins1 and 2

### 3.1. Recruitment of Skeletal Muscle Junctional Proteins

JPH1 and JPH2, and specifically a region in the two junctophilins spanning approximatively from the second half of the joining domain to the first half of the putative α-helical domain, co-immunoprecipitate with Ca_V_1.1 [44]. A 20-residue sequence in the C-terminal domain of Ca_V_1.1 is directly involved in the interactions with junctophilins and in the recruitment of Ca_V_1.1 to triads [45]. Additionally, Ca_V_1.1 is recruited to junctions formed by JPH2 when the proteins are expressed in non-muscle cell models together with the Ca_V_1.1 auxiliary subunit β1a and Stac3 [46]. Overall, this indicates that JPH1 and JPH2 have an active role in recruiting the voltage-gated Ca^2+^ channel to triads by binding directly to the channel; the disruption of this interaction interferes with the assembly of the triad [44,45].

JPH1 co-immunoprecipitates with RyR1, a behavior that has not been observed for JPH2 [47]. Nonetheless, JPH1 KO mice can still perform EC coupling [48], suggesting that the presence of RyR1 in junctions does not depend solely on JPH1. Therefore, the recruitment of RyR1 in JPH2-induced junctions might be due to a weak interaction that is not detected in biochemical assays or requires the presence of additional proteins, with Ca_V_1.1 being a likely candidate. Interestingly, a 28-residue region involved in the direct interaction with the cytoplasmic domain of RyR1 was identified in the divergent domain of the neuronal isoform JPH3 [49], but no homologous sequences have been found in either JPH1 or JPH2, suggesting that different junctophilins might employ different strategies to recruit RyR1 to junctions.

### 3.2. Recruitment of Cardiac Muscle Junctional Proteins

The cardiac L-type Ca^2+^ channel Ca_V_1.2 co-immunoprecipitates with JPH2 [50,51], indicating that, as in skeletal muscle, JPH2 likely plays a role in recruiting the voltage-sensor channel in the dyads and peripheral couplings of cardiac muscle. Notably, the same C-terminal sequence identified as the Ca_V_1.1 site of interaction with junctophilins, is conserved in Ca_V_1.2 [45], suggesting that this sequence might also be involved in Ca_V_1.2–JPH2 interactions.

Differently from what was found with JPH2 and RyR1, co-immunoprecipitation was observed between JPH2 and RyR2 [50,51], indicating a stronger interaction between the two proteins. This interaction is disrupted by the E169K substitution located towards the N-terminal end of the JPH2 joining domain [52] and weakened by the R420Q mutation in RyR2 [53]. A stronger interaction with RyR2 might be required by JPH2 because it is the only junctophilin isoform expressed in cardiac muscle and possibly because cardiac muscle lacks the additional stabilization provided by the mechanical connection between RyR1 and Ca_V_1.1 that exists in skeletal muscle [15].

## 4. Functional Studies on Junctophilins 1 and 2

### 4.1. Junctophilin 1

JPH1 knock-out mice die within 24 h after birth due to suckling defects leading to undernourishment. The suckling defect is likely due to muscle weakness since the neuronal suckling reflexes are normal in knock-out mice [48]. Functional studies on isolated hindlimb muscle showed abnormal twitch tension and a greater dependency on extracellular calcium in KO mice muscles, suggesting that a significant fraction of RyR1s in the junctional SR are not directly coupled with the Ca_V_1.1 channels in the T-tubules and therefore operate via calcium-induced calcium release. Nonetheless, knock-out (KO) mice are still relatively mobile and show skeletal muscle-type EC coupling to a certain degree, indicating that JPH2 can support voltage-induced Ca^2+^ release in the absence of JPH1. From a structural point of view, although no major disorganization of the fiber is noticed at the light microscopy level, evident alterations are noticeable at the ultrastructural level [48,54]. In particular, the skeletal muscle of wt and JPH1 KO mice show a similar development in the embryonic stages until shortly after birth. At this age, wt muscle experiences a significant increase in JPH1 expression, which is temporally correlated with the transition from immature SR-PM junctions, mainly organized in dyads at this stage, into fully formed triads. This transition is absent in JPH1 KO muscle [48,54], suggesting that JPH2 is important in forming the dyads, while JPH1 has a crucial role in the conversion from dyads to triads in the fully mature skeletal muscle. The knocking down of junctophilins using sh-RNA, leads to the impairment of store-operated Ca^2+^ entry (SOCE), altered intracellular calcium release and intracellular calcium stores [55] and to a reduction in RyR1 and Ca_V_1.1 co-clustering associated with a decrease in Ca_V_1.1 membrane expression [44] both in myotubes and muscle fibers. In both these studies, a shRNA against both JPH1 and JPH2 was used; hence, it was impossible to distinguish each isoform’s relative contribution to the resulting phenotype. In 2010 Li and collaborators [56] showed that store-operated Ca^2+^ entry was severely impaired in JPH1 KO mice myotubes, suggesting that JPH1 substantially contributes to maintaining an efficient SOCE in skeletal muscle.

### 4.2. Junctophilin 2

JPH2 knock-out mice die in utero due to cardiac failure. Ultrastructural analyses on embryonic myotubes of KO mice revealed a substantial reduction in the number and extension of peripheral couplings [27]. To avoid the complication related to the early mortality of KO mice, van Oort et al. generated a conditional JPH2 knockdown mice model to assess the effect of JPH2-reduced expression in the mature heart [57]. Inducing JPH2 knockdown led to an increased frequency of heart failure events. At the cellular level, this was explained structurally by T-tubule remodeling and destabilization and disorganization of the dyads, and functionally by Ca_V_1.2 and RyR2 uncoupling and the consequent reduction in the efficiency of calcium-induced calcium release. An increase in the frequency of calcium sparks was noticed in knocked down isolated cardiomyocytes, suggesting that JPH2 might also modulate RyR2 by reducing its activity.

A number of point mutations in JPH2 have been discovered in association with hypertrophic cardiomyopathy and atrial fibrillation. The localization of these mutations spans from the N-terminal MORN motifs to the divergent domain at the C-terminus (Figure 2, Table 1), indicating that multiple regions of JPH2 are involved in supporting cardiac muscle structure and function. A subset of all JPH2 mutations associated with cardiomyopathies has been functionally characterized in cardiomyocytes or cardiomyocyte-derived cell lines (Table 1).

Amino acid substitutions N101R and Y141H in the MORN IV and VI, respectively, and S165F in the joining domain, cause similar phenotypes such as JPH2 mislocalization, reduction in spontaneous Ca^2+^ signaling and increased cell size in HL-1 and H9c2 cell lines [59]. Mutations Y141H and S165F were also tested in skeletal muscle myotubes [60], where they were found to induce myocytes hypertrophy, reduce EC coupling gain and increase intracellular Ca^2+^ concentration. Additionally, Y141H but not S165F pathogenically increased store-operated calcium entry [60]. Mutation E169K, located in the joining domain, causes weaker binding between JPH2 and RyR2 and increased spontaneous Ca^2+^ leakage from the SR in the form of a spontaneous Ca^2+^ release and increased Ca^2+^ sparks in isolated cardiomyocytes from a pseudo-knock-in mouse model [52]. The A405S mutation is located in the putative α-helical region of JPH2. The equivalent mutation introduced in mice (A399S) resulted in cardiomyocytes with an irregular T-tubule pattern but otherwise relatively normal Ca^2+^ signaling with only a moderate increase in sarco–endoplasmic reticulum Ca^2+^ ATPase (SERCA) activity.

It is still unclear whether most of these mutations specifically disrupt binding sites crucial for the interaction of JPH2 with other junctional proteins or whether they alter the structure and stability of the protein with detrimental effects on JPH2 folding and trafficking and, consequently, T-tubular remodeling and impaired SR-PM junction formation. Recent work from Gross and collaborators [66] shows reduced co-immunoprecipitation in the plasma membrane fractions of ventricular myocytes between Ca_V_1.2 and a JPH2 mutant in which seven random mutations were introduced in the joining domain. Based on structure prediction simulations, the authors found that the seven mutations did not compromise the overall organization of JPH2. This work supports the possible role of the joining domain in the binding with Ca_V_1.2. However, the mutated JPH2 showed a significantly reduced ability to form dyads in ventricular myocytes, suggesting that the lack of interactions with Ca_V_1.2 is likely not the only consequence of these mutations for JPH2 function.

## 5. Post-Transcriptional Regulation of JPH1 and JPH2

Conditions of elevated cytosolic Ca^2+^ concentrations lead to fragmentation of junctophilins 1 and 2 in skeletal and cardiac muscle. Murphy and colleagues [67] determined that exposure to elevated (≥20 µM) intracellular [Ca^2+^] for 60 min led to the almost complete loss of full-length JPH1 and JPH2 in skeletal muscle fibers. This loss is mirrored by a loss of contractile force in skinned skeletal muscle fibers after just one minute of exposure to 40 μM Ca^2+^. The same authors also observed fragmentation of JPH1 after raising the intracellular [Ca^2+^] by supraphysiological stimulation of the muscle fiber. Interestingly, the proteolysis of JPH1 temporally matched the autolytic activation of calpain-μ (calpain1). The link between calpain, specifically calpain1, and JPH1 cleavage was recently confirmed by data from Tammineni and colleagues in patients with malignant hyperthermia susceptibility (MHS) and muscle cell lines [68]. Patients with MHS carry mutations in the RyR1 (most often) or other proteins involved in EC coupling that cause leakage of Ca^2+^ from the SR and chronic increases in cytoplasmic Ca^2+^ concentration. Tammineni and colleagues observed increased fragmentation of JPH1 in MHS individuals compared to healthy subjects. Furthermore, the fragmented JPH1 abandons the triad, and its C-terminal fragment translocates into nuclei, where it regulates the transcription of genes known to be altered in MHS. Upon identification of a potential calpain1 cleavage site in JPH1, it was confirmed that treatment of human muscle lysate with Calpain1 generated the same fragments of JPH1 observed in MHS individuals, and the addition of a calpain1 inhibitor prevented fragmentation.

After the identification of several putative calpain binding sites also in JPH2 [51,69], the implication of calpain in the proteolytic regulation of JPH2 was verified using calpain inhibitors to rescue the loss of JPH2 in an inducible heart failure mouse model and in mice cardiomyocytes after ischemia/reperfusion [70]. Akin to what was shown by Tammineni and colleagues in skeletal muscle, Guo and collaborators also observed that digestion by calpain1 releases several fragments of JPH2 [51]. Among these fragments, a ~ 75 kDa N-terminal peptide (JPH2-NTP), generated by calpain1 cleavage at residues R565/T566 in the JPH2 divergent region, migrates into the nucleus, where it binds to TATA box regions and interacts with the transcription machinery [69]. Somewhat in contrast with the findings of Guo et al., recent work from Lahiri and colleagues [71] describes the presence of a C-terminal JPH2 fragment (JPH2-CPT), generated by cleavage of full-length JPH2 at residues G482/T493, in the heart of human patients with heart failure (HF) and of mice HF models. Unlike the JPH2-NTP fragment described earlier, JPH2-CTP seems to be specifically generated by calpain2 in vivo. This specificity might be of importance since calpain2 is activated by millimolar cytoplasmic Ca^2+^ concentrations, which are more likely to be reached in pathological conditions, while calpain1 is activated at more physiological (micromolar) Ca^2+^ concentrations. Interestingly, JPH2-CTP shares the same nuclear localization signal sequence as JPH2-NTP and similarly localizes into the nuclei. However, while JPH2-NTP associates with chromatin and acts as a transcription factor, JPH2-CTP is confined in nuclear sub-compartments and does not seem to affect transcription. The nuclear localization of JPH2-NTP and JPH2-CTP leads to opposite functional outcomes. While JPH2-NTP regulates genes involved in calcium homeostasis to possibly protect the cells from the consequences of elevated intracellular calcium, JPH2-CTP nuclear localization is causative of cardiomyocytes hypertrophy, an early marker of pathogenic cardiac remodeling. Altogether, these results indicate a fine modulation of junctophilin 1 and 2 as a way to regulate intracellular Ca^2+^ homeostasis and possibly reduce EC coupling gain in conditions of excessive intracellular Ca^2+^ concentration. The work from Tammineni et al. and Guo et al. also opens the possibility that cleaved junctophilins might serve as transcription modulators to further contribute to the Ca^2+^ regulation of striated muscle. At the same time, the seemingly contradictory results from different authors indicate that further research is still needed to fully elucidate the different pathways and functional implications of the post-transcriptional regulation of junctophilins.

To further add to the complexity of junctophilin post-transcriptional regulation, calpains might not be the only proteins that cleave junctophilins; work from Chan and colleagues [72] showed a protective effect of a metalloproteinase-2 (MMP-2) inhibitor in murine hearts after acute ischemia-reperfusion injury. This effect was attributed to a reduction in JPH2 cleavage by MMP-2. Such cleavage was demonstrated in vitro in mouse heart extracts after incubation with MMP-2 with or without an inhibitor.

## 6. New Insights from Deep Learning Protein Structure Prediction

Remarkable advancements in protein folding prediction were recently achieved by the artificial intelligence software Alphafold2 [73]. Alphafold2 is a giant leap forward in the reliability of protein folding prediction compared to similar existing software [74], and it has already been used to predict the structure of nearly the entire human proteome. Based on Aplhafold2 prediction models, junctophilin 1 and 2 show a similar 3D structure, also shared by the neuronal isoforms JPH3 and JPH4 (UniProt protein ID: Q9HDC5, Q9BR39, Q8WXH2, Q96JJ6 for human JPH1, JPH2, JPH3 and JPH4, respectively). The structure of the most ordered domains of junctophilin, specifically the MORN repeats, the α-helical region and the transmembrane domain, are predicted with high confidence by Alphafold2. In contrast, the joining and divergent domains are likely disordered, at least in the isolated protein, and the structure cannot be predicted with reasonable confidence. According to the prediction model (Figure 3), the MORN domains are arranged in an extended “half-pipe” configuration, with the α-helical domain lying on the convex side of this half-pipe (Figure 3B,C) and establishing interactions with charged residues in the MORN domains (see the zoomed-in region in Figure 3C for an example).

The structure obtained using Alphafold2 is substantially different from what was previously predicted using RaptorX software by Gross and colleagues [66]. In the structure described by Gross et al., the α-helical domain extends beyond the MORN domains without interacting with them at all. However, Alphafold2 software is considered to be more accurate than most (if not all) of the currently existing structure-predicting software, especially for proteins for which no homologous structures exist [75,76], and the reciprocal arrangement of JPH2 MORN motifs and α-helical domain predicted by Alphafold2 agrees with data from Li and collaborators [32] based on the crystal structure of the protein MORN4. MORN4 contains a series of MORN motifs arranged in a half-pipe configuration followed by a brief α-helical region. The helical region stabilizes the MORN domains by lying over part of the convex side of the half-pipe. The structure solved by Li and colleagues is in many ways very similar to the sequence predicted by Alphafold2 for JPHs. Furthermore, in MORN 4, the concave side of the MORN half-pipe structure, containing most of the conserved residues that define the MORN domain, engages in the binding with the α-helical region of myosin3a. It is conceivable that the concave side of the junctophilin MORN motifs could also participate in protein–protein interactions with components of the EC coupling machinery. The particular arrangement of the α-helical domain with respect to the MORN motifs predicted by Alphafold2 and suggested by the observations of Li and colleagues challenges the classic view of the α-helical domain as the spacer that spans most of the junctional gap (see schematic representation in Figure 2) and points to the divergent domain as the region that most likely fulfills this role.

## 7. Closing Remarks

Junctophilin1 and 2 are crucial players in EC coupling and striated muscle physiology. They form and stabilize the specialized ER-PM junctions, which are the functional platforms at which EC coupling is executed and actively participate in the recruitment of crucial components of EC coupling into such junctions. Under particular pathological conditions leading to elevated cytoplasmic Ca^2+^ concentrations, both JPH1 and JPH2 can be potentially utilized by the cell to ameliorate the consequences of such elevated Ca^2+^. Specifically, Ca^2+^-induced cleavage of JPH1 or JPH2 has the dual effect of uncoupling CaV and RyR, reducing calcium release from the SR and producing JPH fragments that traffic to the nucleus and up-regulate or down-regulate specific genes involved in intracellular Ca^2+^ homeostasis. Therefore, it is not surprising that knocking out or knocking down JPH1 and JPH2 severely impairs EC coupling in animal models. While mutations in JPH1 are generally not associated with significant muscle diseases in humans, possibly due to the compensatory effect of the concurrent expression of JPH2 in skeletal muscle, numerous point mutations in JPH2 have been identified in patients with cardiomyopathies. Although the functional effects of some of these mutations have been explored in animal models, it is still, for the most part, unclear whether JPH2 mutations destroy the ability of the protein to interact with other partners or whether they cause JPH2 misfolding or instability. If, on one end, recent progress in protein structure prediction can help to infer the effect of at least some of JPH2 mutation, some disagreement between different structure-predicting software still exists. Solving the protein’s actual structure, preferably in conjunction with its binding partners, would represent a big step forward in investigating and understanding the nature of junctophilins’ interactions with other junctional proteins and the pathological consequences of the disruption of such interactions.

## Figures and Tables

**Figure 1 biomolecules-12-00109-f001:**
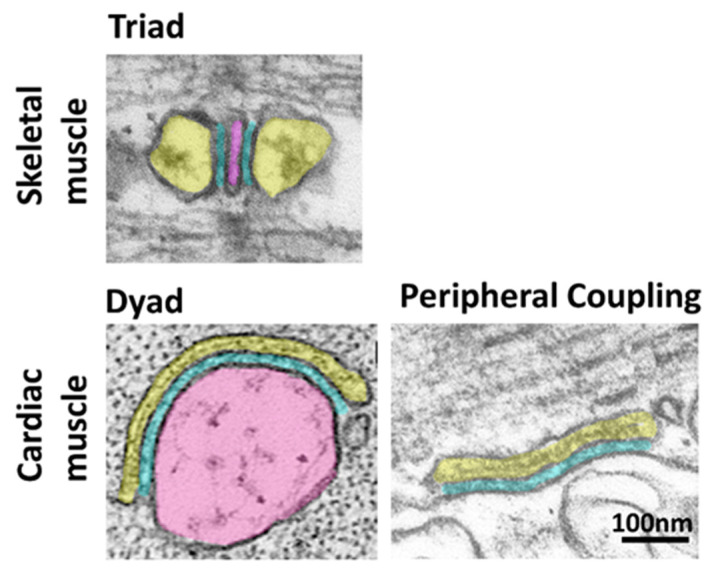
Organization of ER-PM junctions in skeletal and cardiac muscle. Thin section electron micrographs illustrating the different organization of ER-PM junctions in skeletal and cardiac muscle. The sarcoplasmic reticulum and the T-tubule are pseudo-colored in yellow and pink, respectively. The gap separating the T-tubule and the SR membrane is pseudo-colored in blue. The T-tubule is absent in cardiac peripheral couplings since the SR is juxtaposed to the plasma membrane at the periphery of the fiber. Images are from Perni et al. [4] (triad), Lavorato et al. [5] (dyad) and Perni et al. [6] (peripheral coupling).

**Figure 2 biomolecules-12-00109-f002:**
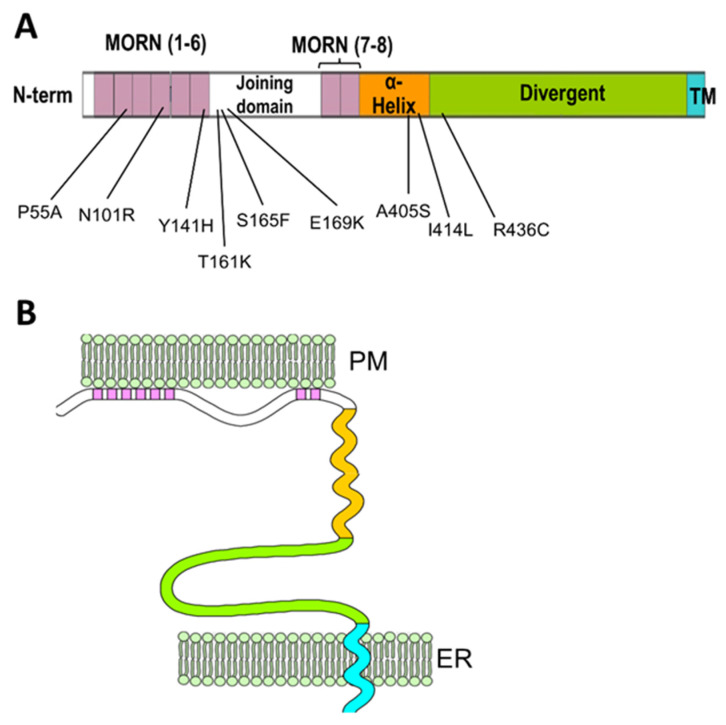
Schematic representation of junctophilin’s domain structure and its arrangement in the ER-PM junction. (**A**) Linear map showing the first (I through IV) and second (VII to VIII) set of MORN domains (in pink) separated by the joining domain (white). The α-helical domain (orange) follows MORN VIII and is separated from the transmembrane (TM, in cyan) by the long divergent domain (green). The numbers underneath the map indicate mutations associated with human cardiomyopathies identified in JPH2 and their relative positions. (**B**) Schematic representation of junctophilin’s organization in the ER-PM junctions (adapted from Garbino et al. [28]). The MORN motifs associated with the internal leaflet of the plasma membrane and the C-terminal transmembrane domain embedded in the endo/sarcoplasmic reticulum membrane allow junctophilins to bridge the two membrane systems together. Different domains are color-coded as indicated in (**A**).

**Figure 3 biomolecules-12-00109-f003:**
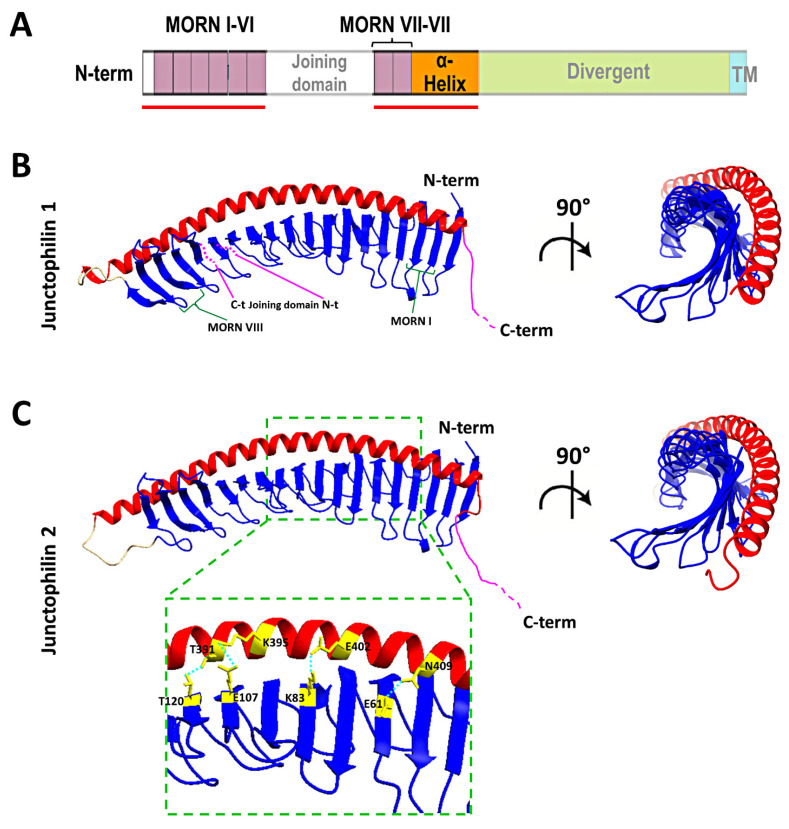
Structure of the MORNs and α-helical domains of human junctophilin1 and junctophilin2. (**A**) Schematic representation of junctophilin domain as shown in Figure 2; the solid red lines indicate the regions for which the structure is predicted with high fidelity by Alphafold2 and illustrated in (**B**,**C**). (**B**,**C**) predicted structures of the MORNs-α-helical domains of junctophilin1 (**B**) and junctophilin2 (**C**). The α-helical domain (in red) lies on the convex side of the MORN domains half-pipe structure (in blue) in both junctophilin1 and junctophilin2. β-sheet hairpins forming MORN domains I and VIII (green parentheses) and the position of the N-terminal end (N-t) and C-terminal end (C-t) of the joining domain (in pink), which is absent in this representation, are indicated in (**B**). The inset in (**C**) shows some of the residues that form the hydrogen bonds that stabilize the association between the MORN domains and the α-helical domain of junctophilin2.

**Table 1 biomolecules-12-00109-t001:** Identified mutations in JPH2 linked to hypertrophic cardiomyopathy (HCM) and atrial fibrillation (AF).

Mutation	Position	Reference	Functional Characterization	Human Phenotype
P55A	MORN II	[58]	Uncharacterized	HCM
N101R	MORN IV	[59]	JPH2 mislocalization, disrupted Ca^2+^ signaling, cardiomyocytes hypertrophy.	HCM
Y141H	MORN VI	[59,60]	JPH2 mislocalization, disrupted Ca^2+^ signaling, cardiomyocytes hypertrophy. In skeletal muscle: abnormal ER-PM junctions, increased store-operated Ca^2+^ entry, decreased EC coupling gain, myotubes hypertrophy.	HCM
T161K	Joining domain	[61]	Uncharacterized	HCM
S165F	Joining domain	[59,62]	JPH2 mislocalization, disrupted Ca^2+^ signaling, cardiomyocytes hypertrophy. In skeletal muscle: Abnormal ER-PM junctions, decreased EC coupling gain, myotubes hypertrophy,	HCM
E169K	Joining domain	[52]	Reduced binding to RyR2, increases spontaneous Ca^2+^ release and Ca^2+^ sparks frequency	AF
A405S	α-helical domain	[52,63]	Irregular T-tubule pattern, mild effect on calcium signaling in the equivalent mutation (A399S) in mice.	HCM
I414L	α-helical domain	[64]	Uncharacterized	Dilated Cardiomyopathy
R436C	Divergent domain	[65]	Uncharacterized	HCM

## Data Availability

Not available.

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
