# Peer review of "The Builders of the Junction: Roles of Junctophilin1 and Junctophilin2 in the Assembly of the Sarcoplasmic Reticulum–Plasma Membrane Junctions in Striated Muscle"

_biomolecules, 2022, doi:10.3390/biom12010109_

Round 1

Reviewer 1 Report

The review is complete and well written. Only minor revision is required:

line 48: please add (SR) after "sarcoplasmic reticulum".

line 72: please replace t-tubule with T-tubule

line 127: please add a comma after "....transmembrane domain)"

line 188: Please add (KO) after "knock.out"

line 209: please replace Knock-out with knock-out

line 241: please add "Reticulum" after "Sarcoplasmic Endoplasmic"

line 307: please replace "panel 2" with "panel B"

Figure 3: Please explain the meaning of the two scissors in panel A.

Reviewer 2 Report

This is a very well-written review on the role of Junctophilins, a class of proteins that can mediate the formation of ER-PM junctions, and which have proven critical in muscle excitation-contraction coupling. The review is very timely, is very clear, and will be a great starting point for anybody new into the field to learn about junctophilins. The author is to be commended to include a wide range of methods that have been used in literature, and to also include the predictions from Alphafold2.

In only have a few minor comments, which could be taken care of very quickly:

1) the section on post-translational modifications covers the cleavage by calpain, and also the observation that an N-terminal fragment can travel to the nucleus, affecting transcription.  It would be great if some of the controversy around the exact calpain cleavage site could be covered. For example, there is the report by Lahiri et al (2020; Basic Res Cardiol) that mentions a different cleavage site and a C-terminal fragment being responsible for the remodeling.

2) The author references the paper by Nakada et al, which suggest an interaction between JPH and a C-terminal peptide in Cav1.1. It's worth mentioning that this peptide sequence is conserved in CaV1.2 as well, which could be included in section 3.2

3) Line 65: the sentence, as written, could be interpreted by novice readers that there are direct interactions between RyRs and CaVs, whereas this still remains to be proven. Although 'interactions' could also be seen as just 'functional' (and indirect), perhaps it's safer to exchange 'the interactions' with 'cross-talk'.

4) Although this focus is on JPH, there is very little mention of STAC3 and CaVbeta1, two other important players in EC-coupling. If space permits, a few lines highlighting their proposed binding sites in CaVs and their importance for EC-coupling , in comparison with JPH, would be great.
